# Orally Bioavailable Proteolysis-Targeting Chimeras: An Innovative Approach in the Golden Era of Discovering Small-Molecule Cancer Drugs

**DOI:** 10.3390/ph17040494

**Published:** 2024-04-12

**Authors:** Rohan Kalyan Rej, Srinivasa Rao Allu, Joyeeta Roy, Ranjan Kumar Acharyya, I. N. Chaithanya Kiran, Yesu Addepalli, V. Dhamodharan

**Affiliations:** 1Rogel Cancer Center, Department of Internal Medicine, University of Michigan, Ann Arbor, MI 48109, USA; srallu@med.umich.edu (S.R.A.); acharyyr@med.umich.edu (R.K.A.); 2Rogel Cancer Center, Department of Medicinal Chemistry, College of Pharmacy, University of Michigan, Ann Arbor, MI 48109, USA; joyeetar@umich.edu; 3Department of Chemistry and Chemical Biology, Harvard University, Cambridge, MA 02139, USA; chaithanyakiranindukurunaga@fas.harvard.edu; 4Department of Biochemistry, University of Texas Southwestern Medical Center, Dallas, TX 75390, USA; addepalli.yesu@utsouthwestern.edu; 5Institute of Organic Chemistry, Center for Nanosystems Chemistry, University of Wuerzburg, Am Hubland, 97074 Würzburg, Germany; dhamodharv@gmail.com

**Keywords:** PROTAC, cancer oral drug

## Abstract

Proteolysis-targeting chimeras (PROTACs) are an emerging therapeutic modality that show promise to open a target space not accessible to conventional small molecules via a degradation-based mechanism. PROTAC degraders, due to their bifunctional nature, which is categorized as ‘beyond the Rule of Five’, have gained attention as a distinctive therapeutic approach for oral administration in clinical settings. However, the development of PROTACs with adequate oral bioavailability remains a significant hurdle, largely due to their large size and less than ideal physical and chemical properties. This review encapsulates the latest advancements in orally delivered PROTACs that have entered clinical evaluation as well as developments highlighted in recent scholarly articles. The insights and methodologies elaborated upon in this review could be instrumental in supporting the discovery and refinement of novel PROTAC degraders aimed at the treatment of various human cancers.

## 1. Introduction

Target protein degradation (TPD) is a promising therapeutic strategy for disease treatment. Proteolysis-targeting chimera (PROTAC) technology is one of the most dynamic therapeutic modalities of TPD, which operates by hijacking the cellular ubiquitin proteasome system (UPS) to induce pathogenic protein degradation. PROTAC molecules consist of a protein-targeting moiety and a ubiquitin ligase moiety connected by a linker. These heterobifunctional entities operate by bringing the target protein into proximity with a ubiquitin ligase, leading to the transfer of ubiquitin and the subsequent degradation of the target protein through the UPS (Figure 1) [1,2,3]. At present, the majority of PROTACs documented in research predominantly target either Mouse double minute 2 homolog (MDM2), inhibitor of apoptosis proteins (IAP), Von Hippel–Lindau tumor suppressor (VHL), or cereblon. Nevertheless, the range of E3 ligases leveraged by PROTACs is steadily growing, as evidenced by the recent inclusion of ligases such as DCAF15/16 [4,5,6,7]. However, the majority of MDM2 or VHL-based molecules failed to achieve significant levels of oral bioavailability, a result that may have been expected due to their considerable molecular weights and the contribution of hydrogen bond donors/acceptors (HBDs/HBAs) from the E3 ligand component. In contrast, cereblon-based molecules present a starting point that is more compatible with oral drug properties owing to their reduced molecular weight, more favorable HBD/HBA profile, and greater lipophilicity [8,9].

## 2. The Rise of PROTAC Technology: A Versatile Game-Changer

The idea of PROTACs was groundbreaking because it imagined that a small molecule could function as a catalytic inhibitor. Put differently, a single PROTAC molecule could prompt a post-translational modification in the target protein (TP), allowing it to inactivate numerous TP molecules [10]. This method, often referred to as “event-driven” inhibition, operates through a transient interaction, contrasting with the constant presence required by traditional “occupancy-driven” inhibitors that rely on forming stable, enduring complexes for TP neutralization [2]. Such a paradigm shift suggested the possibility of effectively targeting proteins that do not have the deep binding pockets standard for high-affinity small molecule interactions. Apart from this, a significant benefit of PROTACs is that they do not need the TP ligand to act as an inhibitor, allowing binding to areas on the TP that may be distant from the functional site. This opens up substantial possibilities for streamlining the development of highly selective degraders for specific proteins within families where the active sites are nearly identical. This sub-stoichiometric degradation allows PROTACs to be effective at low dosages, culminating in a prolonged pharmacodynamic response with low risks of adverse effects. Moreover, the advantages of transient binding mode and the low binding affinity requirement make it possible to turn “undruggable” proteins into “druggable” targets [11,12,13]. This is especially relevant in diseases where achieving functional activity with small molecules is challenging, such as those involving traditionally undruggable targets, proteins with scaffolding functions, and protein aggregates. Additionally, PROTACs show promise in diseases where drug resistance is linked to target protein overexpression or mutation as well as situations requiring isoform selectivity [1,2]. Therefore, over the past decade, PROTACs have gained tremendous momentum from both academia and the pharmaceutical industry in exploring the potential of oral PROTAC degraders as innovative therapeutic agents. Figure 2 shows the development timeline of PROTAC degraders in clinical trials [14].

## 3. Oral Protein Degraders: Progress and Limitations

In 2001, PROTACs were first proposed by Craig M. Crews and Raymond J. Deshaies for proof-of-concept validation [3]. They have now become an established modality in drug discovery, with the first examples reaching the clinic (ClinicalTrials.gov Identifiers NCT03888612 and NCT04072952) [15]. Even though there is increasing curiosity in the use of PROTACs as a treatment option and a surge in related scholarly articles, information regarding their absorption, distribution, metabolism, and excretion (ADME) characteristics is just starting to surface. Their impact on drug development was initially limited for several years due to the absence of suitable drug-like ligands for E3 ubiquitin ligases, as the first PROTACs used peptide ligands resembling natural degrons [16,17]. The landscape shifted with the creation of a synthetic, cell-permeable peptidomimetic ligand for the VHL E3 ubiquitin ligase, paving the way for the development of cell-permeable PROTACs. Another pivotal breakthrough came with the realization that small molecules such as thalidomide and lenalidomide function as ligands for the cereblon E3 ubiquitin ligase [18,19]. Following the emergence of these bioavailable E3 ubiquitin ligase ligands, there has been a surge in the generation of PROTACs incorporating these entities, targeting a wide range of target proteins (TPs). This surge in activity has recently culminated in the development of the inaugural clinical candidates by different biotech firms, showcasing the advances in PROTAC-based therapeutic applications (Figure 2).

Presently, as we experience a revival in the field of small molecule treatments, including those involving PROTACs, the once well-regarded Lipinski’s Rule of Five increasingly seems like a relic from a past era. The oral route of drug administration remains the most favored due to its convenience and the ability to easily modify the dosage and timing. The unique property of degrader molecules, especially those aimed at proteins with prolonged half-lives, lies in their capacity to achieve lasting pharmacodynamic effects. This sustained impact is often due to the slow clearance of the drug from the target site. For optimal efficacy, it is essential to ensure that a sufficient level of the drug is delivered and sustained at the site of action, even if the protein degradation does not remain constant throughout. However, the development of PROTACs faces challenges due to their large size and poor physicochemical properties, which result in a low degree of absorption and undesirable pharmacokinetic properties when administered orally. These properties are categorized as beyond the Rule of Five (bRo5) and include high lipophilicity, a high molecular weight (MW), a large number of rotatable bonds (RBs), a large polar surface area (TPSA), and high hydrogen bond acceptor (HBA) and hydrogen bond donor (HBD) counts. These limitations negatively affect the drug’s solubility, permeability, and metabolic stability, ultimately affecting its oral bioavailability. For instance, a BET PROTAC that combines JQ1, a BET protein inhibitor, and VHL exhibited a cell permeability 165,000 times lower than that of JQ1 alone. Although this may be a particularly stark contrast, and the catalytic efficacy of PROTACs can to some extent compensate for their permeability deficits, there is nonetheless a strong drive within the field to enhance the pharmacokinetic profiles of PROTAC molecules [20]. The higher lipophilicity of PROTACs has also introduced complications for the standard ADME assay protocols. Traditional cell monolayer permeability tests, like the Caco-2 assays, proved to be largely ineffective due to solubility problems in the test buffers and non-specific binding, which led to poor recovery rates and skewed readings of apparent permeability (Papp) rates [21]. To circumvent these issues, alternative methods such as the Parallel Artificial Membrane Permeability Assay (PAMPA) have been considered due to their adjustable assay conditions, which include variables such as incubation duration, pH levels, and the incorporation of biologically relevant solubility systems. However, even with these modifications, recent findings involving a series of androgen receptor PROTACs showed poor permeability in these alternative systems and brought to light persistent concerns regarding sample recovery [22,23]. Accurately measuring the fraction unbound for highly lipophilic PROTACs has been challenging. A notable finding from our routine plasma protein binding (PPB) assays, which utilize the Rapid Equilibrium Dialysis method, is the frequent inability to retrieve satisfactory compound samples. Upon closer examination, it was discovered that the primary difficulty stemmed from the PROTAC molecules adhering to the cellulose membrane, resulting in the retention of a significant portion of the compound that was initially introduced into the incubator [24,25]. Considering the significant link between lipophilicity and the inhibition of cytochrome P450 (CYP) enzymes, it has become imperative to closely examine the potential for drug–drug interactions (DDIs) when evaluating PROTACs [26]. Despite these challenges, there have been recent advances in the development of orally bioavailable PROTACs (Table 1), and this review aims to provide a comprehensive overview of the strategies used to achieve this goal [27].

## 4. Several Ways to Improve PROTACs’ Oral Bioavailability

In recent years, drug delivery systems have undergone remarkable advancements in terms of drug solubility, permeability, and toxicity reductions. To tackle the limitations of PROTACs, various drug delivery vehicles have been proposed in order to enhance their ADME profiles and ensure that they reach their intended sites of action [28,29]. Formulation techniques are being innovated to improve the cellular uptake and solubility of PROTACs and realize their therapeutic potential. Researchers have introduced structural flexibility into PROTAC molecules by enabling intramolecular hydrogen bond (IMHB) formation, resulting in a more soluble and permeable form [30]. Proteolysis-targeting chimeras synthesized inside cells through click chemistry (CLIPTACs) were effectively employed to break down two crucial targets in cancer therapy (BRD4 and ERK1/2) [31]. To selectively and effectively degrade challenging target proteins, PROTACs have been engineered with specific peptides (p-PROTACs) [32]. Various delivery systems, such as self-emulsifying drug delivery systems and polymeric nanoparticles, have been employed to enhance the solubility of drugs. As an example, Rathod et al. developed an ARV-825-enriched self-emulsifying drug delivery system that forms nano globules (ARVSNEP) in aqueous media, which remarkably increased the solubility of ARV-825 by ~15-fold and showed potential for treating vemurafenib-resistant melanoma [33]. Furthermore, Chen et al. showed that embedding PROTACs and related E3-ligase proteins within a lipid nanoparticle system boosted the permeability of PROTAC molecules and enabled them to escape endosomal and lysosomal degradation post-endocytosis [34]. All these innovative and targeted formulation methods hold significant promise for enhancing the efficacy of drug delivery and action.

Taking food with PROTACs is also considered to be a method for boosting their oral bioavailability, as they tend to have low solubility in water. Recent findings indicate that the solubility of PROTACs improves when using buffers that simulate conditions in the human gut, notably when food is present. Solubility tests in biologically relevant systems like FaSSIF/FeSSIF revealed substantial improvements (>50-fold in some instances), suggesting that such techniques should be more widely implemented [35]. This observation is supported by clinical trial practices that often indicate “once daily” administration of certain oral PROTACs along with food. The ongoing ARV-110 clinical trial (NCT03888612), for instance, specifies that participants should take the medication with meals.

Choosing a compact ubiquitin ligase (UL) moiety is key in determining the properties of PROTACs. Smaller CRBNs are preferred for oral use, and the discovery of ligands with even lower molecular weights is expected in the future (Figure 3).

Selecting an appropriate protein-targeting moiety is critical for the development of orally bioavailable PROTACs. The challenge lies in finding protein-targeting moieties that exhibit a balance of characteristics, such as having fewer hydrogen bond donors and acceptors, fewer rotatable bonds, a smaller polar surface area, and a lower molecular weight. These properties are essential for designing effective oral PROTACs.

Boosting the metabolic stability of PROTACs is a crucial step to protect them from “first pass” metabolism in the liver or intestine, thereby improving oral bioavailability. Strategies include modifying the linker length, adjusting the linker’s anchor point, employing rigid spiro linkers, and altering the linker attachment site.

Enhancing cellular permeability also plays a pivotal role in improving the oral bioavailability of PROTACs by promoting their ability to penetrate target cells and navigate across the intestinal membrane barrier effectively.

The creation of intramolecular hydrogen bonds in PROTACs can enhance their cell permeability by reducing the molecular size and polarity.

Lastly, employing a prodrug strategy can enhance the oral bioavailability of a drug. For example, attaching a lipophilic group to the CRBN ligand in a PROTAC compound to create a prodrug has demonstrated an improvement in PROTAC bioavailability.

## 5. Recent Advancements in Orally Bioavailable PROTACs

### 5.1. Discovery of Orally Bioavailable AR PROTAC Degraders for the Treatment of Metastatic Castration-Resistant Prostate Cancer (mCRPC)

The androgen receptor (AR) plays a crucial role as a nuclear transcription factor necessary for prostate cell growth, survival, and multiplication. It plays a pivotal role in the initiation and progression of human prostate cancer and a subset of human breast cancers [36,37,38]. Androgen deprivation therapy (ADT), achieved through surgical castration or drugs that inhibit androgen production, has been successful in treating advanced and metastatic androgen-dependent prostate cancer [39]. However, prostate cancer often advances to a stage known as castration-resistant prostate cancer (CRPC) after several years of castration therapy [40]. Metastatic castration-resistant prostate cancer (mCRPC) remains incurable and fatal. AR antagonists, such as enzalutamide, apalutamide, and darolutamide, can effectively manage mCRPC [41,42]. Despite this, patients frequently develop resistance to these treatments. In most AR antagonist-resistant tumors, the AR signaling is still active, promoting tumor growth and progression [43]. Typical resistance mechanisms to AR antagonists include amplification and mutation of the AR gene and the expression of AR variants [44]. Therefore, in the last few years, new therapeutic strategies like PROTAC technology have gained considerable momentum for discovering and developing new therapeutic agents [45,46].

As stated above, a typical PROTAC degrader has an MW above 700 and falls outside of Lipinski’s “Rule of Five”, which complicates the attainment of oral bioavailability. Nevertheless, through rigorous optimization, orally administered AR PROTAC degraders have been successfully developed, exemplified by the compounds shown in Figure 4.

Bavdegalutamide (ARV-110) stands out as the first clinically evaluated orally bioavailable AR PROTAC degrader, and its structure was presented at the 2021 AACR national meeting [47,48]. The strategic approach employed in the discovery of bavdegalutamide (**1**) is outlined in Figure 5. The effort was initiated by evaluating various AR ligands and E3 ligands and selecting the reported AR inhibitor [49] from Pfizer in combination with thalidomide as the cereblon ligand, which yielded degrader **16** employing a flexible linker. Compound **16** exhibited a DC_50_ ranging from 1 to 10 nM and a D_max_ level < 50% in the LNCaP cell line. Despite showing promise with oral bioavailability, compound **16** was observed to have a high clearance rate in pharmacokinetic studies. Subsequent refinement of the linker in **16** resulted in compound **17**, which demonstrated a DC_50_ value of less than 1 nM and a D_max_ of over 50%. However, compound **17** exhibited suboptimal in vivo efficacy and possessed a high melting point, indicating potential formulation difficulties. By imposing a conformational constraint on the linker in **17**, compound **18** was obtained, maintaining a DC_50_ value of <1 nM and a D_max_ > 50%. The development process progressed by substituting the AR ligand in compound **18** with an alternative AR antagonist and adding a fluorine atom to the thalidomide component, culminating in the discovery of bavdegalutamide (ARV-110, **1**) as a clinical candidate (Figure 5).

In the LNCaP cell line, bavdegalutamide has a DC_50_ value of 0.24 nM and achieves a D_max_ of 82%. This potent degradative capability is further evidenced by its efficient reduction of androgen receptor (AR) levels by more than 95% in the original VCaP cell line, VCaP cells with AR mutations (including F877L, T878A, M897V, and H875Y), and a VCaP cell line resistant to enzalutamide. Additionally, bavdegalutamide effectively inhibits the production of prostate-specific antigen (PSA). Oral administration of bavdegalutamide is effective at diminishing AR protein levels in VCaP and LNCaP xenograft tumor tissues. Additionally, bavdegalutamide has demonstrated superior inhibition of tumor growth compared with enzalutamide in various AR-positive prostate cancer xenograft models in mice, solidifying its effectiveness [48,50,51,52].

Although bavdegalutamide has a high molecular weight (812), it is still able to achieve sufficient oral bioavailability in mice, rats, and non-rodent species, lending support to its oral use in preclinical safety studies and human clinical trials. Furthermore, it has successfully validated the therapeutic principle of a PROTAC degrader targeting the androgen receptor in metastatic castration-resistant prostate cancer (mCRPC). Recent Phase 2 clinical data have revealed that patients with AR 878/875 mutations have a radiographic progression-free survival (rPFS) of 11.1 months when treated with this drug. Additionally, the drug’s side effect profile has been deemed manageable and appropriate for patients suffering from mCRPC [53,54].

Advancing its prostate cancer pipeline, Arvinas is currently exploring the potential of ARV-766, a new orally available PROTAC molecule (Figure 6). ARV-766 is a novel, orally administered PROTAC molecule under investigation for its ability to target and degrade various resistance-inducing point mutations in the androgen receptor (AR), including the L702H mutation.

The discovery of ARV-766 was presented at AACR 2023. Accordingly, replacement of the six-membered ring in the POI moiety of ARV-110 (**1**) with a tetramethyl cyclobutane ring resulted in the compound (**19**) with the desired genotype coverage. Following this modification, the switch of a chlorine atom for a methoxy group within the AR binder resulted in compound **20** with enhanced oral bioavailability. With a highly effective target protein binder available, modifications to the CRBN component incorporating precise stereochemistry culminated in the discovery of Luxdegalutamide (ARV-766, **2**). Preliminary findings from an ongoing Phase 1/2 clinical study are encouraging, showing that ARV-766 is generally well-tolerated and potentially effective.

According to a recent study, there is a growing frequency of mutations in the ligand-binding domain of the AR, and the incidence of a specific mutation (L702H) is on the rise [55]. Projections indicate that around 11% of mCRPC cases in 2023 will involve the L702H AR mutation (Table 2). Collectively, AR LBD mutations are identified in approximately 20–25% of mCRPC instances.

The reduced effectiveness of bavdegalutamide in treating patients with AR L702H mutation-positive tumors may restrict its clinical benefits for the wider mCRPC patient demographic. Contrastingly, ARV-766 appears to offer enhanced efficacy in patients with AR L702H mutation-bearing tumors relative to bavdegalutamide, as disclosed in Table 3. These findings suggest that ARV-766 is emerging as a promising treatment for individuals with AR mutations associated with resistance to therapy. ARV-766, which represents a new-generation pan-AR degrader, demonstrates an extended efficacy spectrum and a more favorable tolerability profile in comparison with bavdegalutamide, indicating its potential to benefit a broader patient cohort with conditions such as metastatic castration-resistant prostate cancer (mCRPC) and metastatic castration-sensitive prostate cancer (mCSPC). Table 3 shows a profile comparison between ARV-766 and bavdegalutamide [56,57].

Our colleague from the University of Michigan has also reported multiple orally bioavailable and potent AR PROTAC degraders, represented by ARD-2128 (**3**) [58], ARD-2585 (**4**) [59], ARD-2051 (**5**) [60], and ARD-1676 (**6**) [61] (Figure 4).

ARD-2128 was developed based on the structure of ARD-61 (Figure 7) [62,63]. ARD-61 is an extremely potent AR degrader and has also shown effectiveness in enzalutamide-resistant systems and two distinct xenograft tumor models derived from LNCaP and VCaP cells. Despite these successes, ARD-61 was not orally bioavailable in mice. Its lack of oral bioavailability was attributed to factors such as its high molecular weight (MW, 1096), substantial topological polar surface area (TPSA, 189A°), high CLogP value (8.2), and other unfavorable chemical characteristics. To overcome these issues, modifications were made to ARD-61 to enhance its chemical profile. These changes involved shortening the linker and switching the E3 ligase target from VHL to cereblon [58]. The resulting compound, ARD-2128, exhibits DC_50_ values of 0.28 nM in the VCaP cell line and 8.3 nM in the LNCaP cell line. It also demonstrated an impressive pharmacokinetic profile in mice, which included low clearance, a favorable volume of distribution, and 67% oral bioavailability. Oral administration of ARD-2128 effectively lowers AR protein levels and downregulates AR-targeted genes within tumor tissue, significantly inhibiting tumor growth in mouse models.

Figure 8 illustrates the multi-step optimization process that led to the creation of ARD-2585 [59]. First, an AR antagonist known as **21** was conjugated to thalidomide via a linear linker, which facilitated the determination of the most effective linker length. This resulted in the crafting of the potent degrader **22**. Subsequently, linker rigidification using different ring systems was investigated. The final phase encompassed refining the AR antagonist element and meticulously adjusting the rigid linker, culminating in the development of ARD-2585 (**4**). This compound is highly efficacious, achieving DC_50_ values below 0.1 nM in both VCaP and LNCaP cell lines. Additionally, ARD-2585 suppresses cellular proliferation in VCaP and LNCaP cells, with IC_50_ values of 1.5 nM and 16 nM, respectively. It also demonstrates superior pharmacokinetic properties in mice, including 51% oral bioavailability. In comparison with enzalutamide, ARD-2585 is more effective in suppressing tumor growth in the VCaP xenograft tumor model in mice at well-tolerated doses [59].

Based on previous work, another oral AR degrader termed ARD-2051 was also developed by Wang et al. It has become increasingly evident that the oral pharmacokinetics of PROTACs can vary markedly between species, posing challenges for their clinical development [64]. As previously mentioned, ARD-2585 displayed a favorable oral bioavailability of 51% in mice, yet its bioavailability in rats was comparatively lower, at 13%. Therefore, the project aimed to discover a new and potent PROTAC that exhibits excellent oral bioavailability across a broad range of species. The authors hypothesized that decreasing the polar surface area of the target protein ligand might lead to a PROTAC with enhanced oral bioavailability in various animals. Accordingly, compounds **24** and **25** were synthesized with a reduced topological polar surface area (TPSA). In order to improve the binding affinity of the protein of interest, the higher-affinity compounds **26** and **27** were designed [60]. Further analyzing the predicted binding model of the AR ligand, a chiral methyl group was installed onto the 1-position of the spiro ring. This structural modification, along with the subsequent linker refinement, led to the discovery of ARD-2051 (**5**), as depicted in Figure 9.

The in vitro assessments of ARD-2051’s degradation capability revealed that it was several times more effective than ARV-110 at degrading the AR in VCaP and LNCaP cell lines, achieving a DC_50_ value of 0.6 nM for both lines and D_max_ levels of 97% and 92% in VCaP and LNCaP, respectively. Pharmacokinetic studies indicated that ARD-2051 possessed a satisfactory PK profile across multiple species, with oral bioavailability values of 53% in mice, 82% in rats, and 46% in dogs. When evaluating the antitumor effects of ARD-2051 using the VCaP xenograft model in mice, the results showed that the compound significantly inhibited tumor growth, reaching a tumor growth inhibition (TGI) value of 80%, with no toxicity observed at an oral dose of 30 mg/kg.

ARD-1676 (**6**) is another highly potent and orally efficacious PROTAC degrader of the androgen receptor (Figure 10) [61]. This molecule was crafted using AR ligands from the same category as those in ARD-2051 and incorporates a novel cereblon-binding molecule, TX-16. Regarding its degrading capability, ARD-1676 displays DC_50_ values of 0.1 nM in AR-positive VCaP cells and 1.1 nM in LNCaP cells. Its inhibitory effects show IC_50_ values of 11.5 nM in VCaP cells and 2.8 nM in LNCaP cells. ARD-1676 effectively induces the degradation of a broad panel of clinically relevant AR mutants. ARD-1676 has an oral bioavailability of 67% in mice, 44% in rats, 31% in dogs, and 99% in monkeys. Oral administration of ARD-1676 effectively reduces the AR protein level in the VCaP tumor tissue in mice. It inhibits tumor growth in the VCaP mouse xenograft tumor model without any sign of toxicity. Given these properties, ARD-1676 is viewed as an exceptionally promising candidate for the development of treatments for AR-positive human prostate cancer [61]. In summary, this medicinal chemistry effort demonstrated that substantial discrepancies in PK profiles among different species could be mitigated through systematic structural optimization, which included strategies such as reducing the polar surface area and making targeted adjustments to the protein ligand, linker, and E3 ligase components [65].

Qin and colleagues recently reported on a promising orally available PROTAC, BWA-522 (**7**), which has shown efficacy in promoting the degradation of both the full-length androgen receptor (AR-FL) and the AR-V7 variant (Figure 11) [66]. BWA-522 was developed from Ralaniten, a potent AR N-terminal transcriptional domain (AR-NTD) antagonist. By introducing rigid links between Ralaniten and Thalidomide, they were able to form a stable and efficient ternary complex, which led to an enhanced ability to degrade AR-V7 in the synthesized compounds. Accordingly, a series of degraders with rigid linkers, such as four- or six-membered heterocycles, was designed in order to constrain the conformation of the PROTACs. Within this series, BWA-522 stood out with its substantial ability to degrade AR-FL and AR-V7, achieving reductions of 52.4 and 73.1% for AR-FL and 77.3 and 84.6% for AR-V7 at concentrations of 1 and 5 μM in VCaP cells, respectively. Notably, BWA-522 displayed impressive oral plasma levels with a Cmax of 376 ng/mL and an AUC of 5947 h·ng/mL when administered orally at 10 mg/kg in mice, resulting in an oral bioavailability of 40.5%. The pharmacokinetics of BWA-522 were further validated in beagle dogs, showcasing an excellent oral bioavailability of 69% following a 5 mg/kg dose. In an encouraging development, the evaluation of BWA-522 using the LNCaP xenograft mouse model revealed a tumor growth inhibition (TGI) value of 76% at an oral dose of 60 mg/kg, marking a more efficient tumor suppression rate than Ralaniten at 180 mg/kg. Collectively, these findings indicate that BWA-522 is an orally bioavailable degrader targeting the AR-NTD with the potential for clinical application in prostate cancer therapy.

### 5.2. Discovery of Orally Bioavailable ER PROTAC Degraders for the Treatment of ER+/HER2− Advanced Breast Cancer

Endocrine therapy is typically effective in treating the majority of patients with hormone receptor-positive (HR+) advanced breast cancer. However, as time passes, cancer cells become resistant to this therapy by acquiring new mutations or losing hormone receptor expression [67]. This makes them insensitive to standard hormonal therapy inhibitors or cyclin-dependent kinase 4 and 6 (CDK4/6) inhibitors. Researchers are addressing this challenge by developing new therapies to treat the growing number of patients whose disease progresses after effective treatment with CDK4/6 inhibitors. ARV-471 is the first ER PROTAC that progressed into clinical development for treating patients with locally advanced or metastatic ER+ BC (Figure 12) [47,68]. The latest findings from a Phase I/II trial (NCT04072952) revealed that Vepdegestrant is an effective treatment option for patients with ER+, HER2 advanced breast cancer [69,70,71,72].

ARV-471 is a CRBN-based degrader featuring a rigid linker and lasofoxifene, a selective estrogen receptor modulator. The degradation efficiency depends on the actual stereochemistry of the protein-targeting ligand, and the (*S*)-configuration of the cereblon moiety yields a more potent PROTAC molecule than the (*R*)-configuration. ARV-471 demonstrated strong ER degradation with a DC_50_ value of 1.8 nM in MCF7 cells. Preclinical animal studies show that oral administration QD of ARV-471 at 3, 10, or 30 mg/kg leads to significant antitumor activity in xenograft models. Preclinical in vivo studies also showcased that a daily oral dose of ARV-471 induced a significant reduction in tumor size, with a 99% decrease observed at a dose of 10 mg/kg in an *ESR1* mutant PDX model. Pharmacokinetic studies revealed that the dosage of ARV-471 correlated with the extent of exposure, far exceeding the preclinical efficacy benchmarks. Oral administration of 30 mg/kg resulted in an area under the curve (AUC) level reaching as high as 5717 ng × h/mL, along with a substantial half-life (T_1/2_) of 28 h [73]. Therefore, given its status as one of the most developed PROTAC molecules, there is significant optimism that ARV-471 will secure FDA approval to treat a specific group of breast cancer patients.

In 2023, Wang et al. developed ERD-3111, an orally bioavailable estrogen receptor-degrading PROTAC (Figure 13) [74]. The development of ERD-3111 is based on extensive optimization of the linker and ER inhibitor core. The combination of Lasofoxifene and the thalidomide-based orally bioavailable cereblon ligand TX-16 connected through a 6,6-spiro linker produced ERD-1173, which demonstrated a decent estrogen receptor degradation capacity (DC_50_, 5.5 nM; D_max_, 90%) and showcased good oral bioavailability in mice (35%) but only moderate bioavailability in rats (13%). Optimization of the ER antagonist resulted in a 10-fold improvement in the degradation potency of ERD-857 at the expense of oral bioavailability in rats. Next, the ER antagonist core was swapped with the AZD-9833 core to yield ERD-3237, which preserved the DC_50_ while improving the maximum degradation level to 107% and slightly enhancing the oral bioavailability in rats to 14%. The final iteration involved modifying the indazole inhibitor core of ERD-3237. By substituting the pyridine ring with a more electron-deficient difluoro phenyl ring and changing a CF_3_ group to a CHF_2_ group, ERD-3111 was obtained. These adjustments increased the overall hydrophobicity and reduced the intravenous clearance, resulting in an oral bioavailability of 42% in mice, 20% in rats, and an impressive 66% in dogs while delivering an excellent pharmacokinetic profile across these species.

### 5.3. Discovery of Orally Bioavailable BRAF Degraders

Over 300 different missense mutations in BRAF have been discovered in patients, but today’s authorized drugs only address the V600 variants. In addition, resistance tends to develop over time, mainly because of RAF alterations that stop BRAF V600 inhibitors from being effective. Consequently, it is necessary to develop novel treatments that target other mechanisms of activated BRAF [75]. CFT1946 is an orally administered degrader specifically targeting the BRAFV600E mutant protein, and it functions via the recruitment of the CRBN E3 ligase [76,77]. During preclinical studies, CFT1946 distinctly targeted and degraded the BRAF V600E mutation with a DC_50_ of 14 nM in a melanoma cell line containing roughly 9000 proteins. Notably, when combined with the trametinib treatment, CFT1946 led to a more pronounced reduction in ERK phosphorylation and tumor growth inhibition than either CFT1946 alone, trametinib alone, or encorafenib monotherapy [78].

The structural characteristics and in vivo pharmacokinetic (PK) details of CFT1946 were recently revealed at AACR 2023 (Figure 14). CFT1946 incorporates a unique E3 ligase-binding component, achiral dihydrouracil, which avoids the potential for the spontaneous racemization commonly seen in the glutarimide moiety of classic CRBN ligands. The degrader also possesses a rigid linker with a piperidine structure that favors a conformational collapse, resulting in a reduced solvent-accessible surface area (SASA). These innovative structural elements combine to endow CFT1946 with exceptional oral bioavailability, demonstrated by an oral bioavailability (F%) of 89% in rats, while maintaining precise selectivity for degrading BRAF V600E. This showcases a stellar instance of how high oral bioavailability for heterobifunctional PROTACs that exceed the conventional ‘Rule of Five’ can be achieved through a strategic medicinal chemistry initiative, which includes a thoughtful design of the linker component [65].

### 5.4. Discovery of Orally Bioavailable BRD9 Degraders

The protein Bromodomain-containing protein 9 (BRD9), a crucial part of the ncBAF variant of the SWI–SNF chromatin remodeling complex, has been identified as a potential therapeutic target in a subset of sarcomas and leukemias [79]. Overexpression of BRD9 has been observed in various cancers, including non-small-cell lung cancer, hepatocellular carcinoma, pediatric rhabdoid tumors, prostate cancer, and cervical cancer, making it a compelling target for the development of a new therapy for the treatment of a number of human cancers [80,81]. Furthermore, recent studies indicate that tumors deficient in SMARCB1, like synovial sarcoma and malignant rhabdoid tumors, exhibit a synthetic lethal dependency on BRD9 [82]. CFT8634 is a highly selective BRD9 PROTAC currently being investigated in a Phase I clinical trial (Figure 15) [83,84].

The structure of CFT8634, featuring a novel E3 ligase ligand, has been made public, and the process of its development is illustrated in Figure 15. Preclinical studies have shown that CFT8634 is both potent and selective in degrading BRD9 in synovial sarcoma and SMARCB-1-deficient cases, demonstrating a DC_50_ value of 2.7 nM. Additionally, CFT8634 has shown sustained tumor shrinking even after treatment cessation. Importantly, CFT8634 possesses excellent oral bioavailability (F = 83%), a Cl obs of 22 mL/min/kg in rats, and low hERG inhibition (>30 μM).

### 5.5. Oral PROTAC SMARCA2 Degraders

The disruption of epigenetic regulation is a widespread and notable characteristic found in the vast majority of human cancers [85]. A growing body of evidence suggests that targeting epigenetic mechanisms may be an effective strategy for treating, among other diseases, human cancers. The SWI–SNF chromatin-remodeling complex is instrumental in the repositioning of nucleosomes, thereby managing key cellular processes such as transcription, DNA replication, and repair. Approximately 20% of human cancers have been found to harbor mutations that affect the function of chromosome dynamics within this complex [86]. Mutations of SMARCA4 have been identified in a range of cancer types, including ovarian cancer, melanoma, and non-small-cell lung cancer, with an occurrence rate sitting at around 11% across various human cancers. Experiments using RNA interference to knock down SMARCA2 in cells lacking SMARCA4 have shown that these SMARCA4-deficient cells are highly reliant on SMARCA2 for their survival in vitro and in vivo, revealing a relationship of synthetic lethality between the two subunits. Such discoveries powerfully support the potential of SMARCA2 as a valuable target for developing therapies to treat cancers deficient in SMARCA4 [87,88,89].

One rare example of an orally bioavailable PROTAC that utilizes a VHL E3 ligase ligand is ACBI2. In 2022, research by the Farnaby group unveiled structural details of the orally bioavailable SMARCA2 degrader ACBI2 (Figure 16). Their research began with the discovery of compound 12, which had a relatively low oral bioavailability of 4.3% at a dose of 30 mg/kg PO. Remarkably, by incorporating a chiral methyl group into the molecule’s linker portion, they significantly improved the oral bioavailability to 22% in the resultant compound ACBI2. This improvement highlights that **32** (PROTACs based on VHL E3 ligase ligands with suboptimal oral bioavailability) can be substantially improved through deliberate modifications, particularly to the linker segment. Future research is anticipated to concentrate on fine-tuning such linker modifications to further bolster oral uptake [90].

### 5.6. Discovery of Orally Bioavailable BTK Degraders

Bruton’s tyrosine kinase (BTK), a member of the TEC family of kinases, is an essential effector of B-cell receptor (BCR) signaling. Chronic activation of BTK-mediated BCR signaling is a hallmark of many hematological malignancies, which makes it an attractive therapeutic target. Pharmacological inhibition of BTK enzymatic function is now a well-proven strategy for treating patients with these malignancies. In 2024, Nurix Therapeutics published NX-2127 as their development candidate for BTK degradation for treating patients with B-cell malignancies (Figure 17) [91,92]. NX-2127 is a BTK degrader with concomitant immunomodulatory activity. NX-2127 degrades wild-type BTK and ibrutinib-resistant C481S mutant BTK (BTKC481S) in TMD8 cells with a DC_50_ of 4.5 and 31 nM, respectively. NX-2127 also degrades both IKZF1 and IKZF3 with a potency of 57 and 36 nM, respectively, which is similar to the approved drug pomalidomide [93]. The development process of NX-2127 is depicted in Figure 17, showcasing how the molecule evolved through the systematic refinement of its inhibitors, linkers, and cereblon ligands. The linear linker-containing degrader (**13**) showed no oral exposure in mice, whereas rigidification of the linker resulted in NX-0942 showing moderate oral bioavailability (5.4%) in mice. The reduction in the molecular weight by removing the urea moiety from the inhibitor resulted in NX-2127 showing a decrease in DC_50_ by 9-fold and a decrease in D_max_ by 10 units but a significant improvement in oral bioavailability in mice up to 36%. NX-2127 showed an excellent to moderate pharmacokinetic profile across species of 36, 7.1, 1, and 1.2% in mice, rats, dogs, and cynomolgus monkeys, respectively. Single oral doses of 0.3, 3, 10, and 30 mg/kg of NX-2127 in mice reduced BTK levels by 19%, 64%, 79%, and 88%, respectively, in circulating B cells after 24 h. Oral doses of 10 mg/kg in dogs and cynomolgus monkeys reduced BTK levels by 83% and 91%, respectively. In a Phase 1 human clinical trial, CLL patients were treated with 100 mg of NX-2127 orally once daily [94].

NX-5948 is another selective BTK degrader developed by Nurix Therapeutics [95,96]. NX-5948 demonstrated potent degradation capabilities with a DC_50_ of 0.32 nM and 0.21 nM in wild-type and C481S TMD8 cell lines, respectively, both achieving a D_max_ value of 97%. In primary human B cells, the compound boasts a DC_50_ of 0.034 nM with a 98% D_max_. Preclinical studies indicate that NX-5948 is able to cross the blood–brain barrier and effectively degrade BTK in both microglial cells and lymphoma cells residing in the brain. Notably, NX-5948 does not cause significant degradation of the IKZF1/3 proteins in primary human T cells. The journey to developing NX-5948 is illustrated in Figure 17, where it is highlighted that NX-5948 was derived from NX-0942 after a thorough optimization of cereblon ligands. In terms of pharmacokinetics, NX-5948 shows acceptable clearance rates and good oral availability at a 10 mg/kg dosing level in various species. Specifically, after oral administration, the compound exhibited an oral bioavailability of 25% in mice, 16% in rats, 9% in dogs, and 2% in cynomolgus monkeys, ranging from excellent to moderate levels.

### 5.7. Discovery of Orally Bioavailable EGFR Degraders

Epidermal growth factor receptor (EGFR), a transmembrane glycoprotein critical in various signaling pathways involved in cell proliferation, angiogenesis, and apoptosis, is a valuable oncogenic drug target. Small-molecule tyrosine kinase inhibitors (TKIs) and monoclonal antibodies (mAbs) have been developed to target the intracellular and extracellular domains of the EGFR. However, EGFR catalytic structural domain alterations and ongoing drug resistance restrict its clinical application [97]. To overcome these issues, EGFR PROTACs are gaining attention as a novel and promising strategy. The EGFR C797S mutation was identified as the primary on-target resistance mechanism to osimertinib in patients with advanced non-small-cell lung cancer (NSCLC) [98]. Currently, no effective treatment options exist for NSCLC patients harboring mutations (Del19/T790M/C797S and L858R/T790M/C797S). The orally bioavailable EGFR PROTAC HJM-561 (Figure 4) can selectively degrade the EGFR C797S-containing triple mutants. HJM-561 potently inhibits the proliferation of Del19/T790M/C797S and L858R/T790M/C797S Ba/F3 cells while sparing cells expressing wild-type EGFR. Oral administration of HJM-561 shows robust antitumor activity in EGFR Del19/T790M/C797S-driven Ba/F3 CDX and PDX models unresponsive to osimertinib treatment [99]. HJM-561 displayed favorable oral PK properties, with a *C*_max_ of 3677.25 ng/mL, an exposure AUC of 1970.2 h × ng/mL, and an oral bioavailability of 62.8% at 10 mg/kg in an EGFR Del19/T790M/C797S-Ba/F3 cell-derived xenograft model. Osimertinib treatment at 10 mg/kg had no effect on the tumor growth. However, the oral administration of HJM-561 resulted in a significant reduction in tumor volume (58% and 84% at 20 mg/kg and 40 mg/kg doses, respectively). Taken together, HJM-561 as a novel EGFR mutant PROTAC demonstrated robust oral antitumor activity in EGFR Del19/T790M/C797S-driven and osimertinib-resistant Ba/F3 CDX and PDX models [99].

### 5.8. Discovery of an Orally Bioavailable CDK 2/4/6 Degrader by Employing a Prodrug Strategy

Cyclin-dependent kinases (CDKs) are essential in controlling cell division and proliferation in eukaryotic cells [100]. These catalytic proteins become active when bound to regulatory proteins known as cyclins. Of the 16 or more cyclins identified in mammals, several pairings, such as Cyclin B and CDK1, Cyclin A and CDK2, Cyclin E and CDK2, Cyclin D and CDK4, and Cyclin D and CDK6, are crucial for proper advancement through the cell cycle. Apart from their primary role in cell cycle regulation, cyclins and CDKs also have additional functions, including gene transcription regulation, DNA repair, cellular differentiation, and induction of apoptosis. Therefore, inhibitors targeting cyclin-dependent kinases have emerged as potent anticancer agents. Specifically, the inhibitors targeting CDK4/6 have shown significant promise in clinical settings for treating breast and other forms of cancer, either as monotherapies or in combination with other drugs. Drugs such as Palbociclib and Ribociclib, for example, have received FDA approval for use in combination with aromatase inhibitors for treating hormone receptor-positive (HR-positive) and human epidermal growth factor receptor 2-negative (HER2-negative) advanced or metastatic breast cancers. However, the efficacy of these CDK4/6 inhibitors may diminish over time due to the development of primary or acquired resistance to these drugs [101,102].

In 2020, Wei et al. published a study on the prodrug-based, orally bioavailable PROTAC **36** that aims to target the degradation of cyclin-dependent kinases (CDK) 2/4/6 through CRBN recruitment (Figure 18) [103]. Ribociclib was used as a target protein binder for its compelling CDK4/6 binding affinity. However, Ribociclib demonstrated a poor inhibitory effect on CDK2. To address this issue, the piperazine ring in Ribociclib was replaced with a methyl ester group yielding compound **35**. Among the synthesized PROTACs, compound **35** was the most effective for the simultaneous degradation of CDK2/4/6, but it had poor oral bioavailability (<1%). To enhance the oral uptake of the compound, a lipophilic group was attached to the N of the glutarimide section in the cereblon ligand moiety using a prodrug strategy. This increased the oral bioavailability of the compound significantly, achieving an F% value of 68%, thus marking the first instance of improved oral bioavailability for OB-PROTACs through a prodrug approach. Consequently, this breakthrough has paved the way to novel avenues in the development of similar compounds.

## 6. Overcoming the Limitations of Lipinski’s Rule of Five: Utilizing Uptake Transporters

Crafting a bivalent drug for oral administration poses a significant challenge compared with creating inhibitor drugs. Therefore, researchers have been focusing on optimizing the drug delivery system, understanding its metabolism, and improving its solubility and ability to cross cellular barriers. Lipinski’s Rule of Five is a useful guideline when considering a compound’s physical and chemical characteristics, but it may not be applicable for drugs designed for non-oral administration or those actively transported into cells by biological transport mechanisms. To highlight this point, below are three case studies that shed light on the exceptions and provide motivation for the creation of bivalent degrader drugs [104]. The first example of an exception to Lipinski’s Rule of Five is the inhibitors of the Hepatitis C virus (HCV) nonstructural protein 5A (NS5A) replication complex (Figure 19). Despite having large molecular sizes, ranging from 738.9 to 1113.2 Da, which might suggest suboptimal oral bioavailability based on the Rule of Five, inhibitors such as ledipasvir, elbasvir, velpatasvir, ombitasvir, and pibrentasvir have been successfully developed for managing chronic HCV. This is due to the inclusion of an L-Pro-L-Val dipeptide segment in these molecules, allowing them to be recognized by cellular transporters and achieving sufficient plasma exposure when administered orally [105,106]. The second example involves the BCL-2/BCL-XL dual inhibitor navitoclax (ABT-263) and the BCL-2-specific inhibitor venetoclax (ABT-199), which are designed to treat patients with relapsed or refractory chronic lymphocytic leukemia (Figure 19). Despite their considerable molecular weights of 974.6 Da for navitoclax and 868.5 Da for venetoclax, both drugs are orally bioavailable. Studies have shown that their effective absorption can be attributed to transport by the intestinal lymphatics, which also contributes to the increased systemic bioavailability by bypassing the first-pass metabolism in the liver [107,108,109]. The third example is the bitopic mTOR inhibitor RapaLink-1, which has a molecular weight of 1784.2 Da (Figure 19). Despite its size, RapaLink-1 has notable in vivo effectiveness, indicating its strong ability to permeate cells. This suggests the existence of a cellular uptake mechanism that facilitates the entry of such large compounds into the cytoplasm. Recent research has revealed that interferon-induced transmembrane proteins (IFITMs) enhance the cellular absorption of these linked bitopic substances. Interestingly, the assistance provided by IFITMs increases with the length of the linker in the compounds. This discovery provides valuable insights for designing other linked chemical structures that might also leverage this uptake mechanism [110,111,112]. In conclusion, taking advantage of transporters in the intestines and brain for the absorption of bivalent degrader drugs may offer a novel approach to overcoming challenges associated with oral bioavailability and reaching into the central nervous system (CNS).

Table 4 summarizes the physiochemical properties of oral PROTAC degraders. For hydrogen bond donors (HBDs), four or less bonds is the regulating threshold for favorable oral bioavailability. Specifically, molecules with three or more HBDs typically possess one or two internally compensated hydrogen bonds. Hence, a more precise delineation would be that only two to three unsatisfied HBDs are acceptable. This insight provides chemists with additional information in designing molecules with HBD counts as long as they ensure that any HBDs beyond two to three are internally neutralized. Similarly, for hydrogen bond acceptors (HBAs), the approximate maximum limit is 15. Specifically, compound CFT1946 with 15 HBAs was successful in meeting this limit [64]. In previous studies, such as those by Veber et al., the proposed cutoffs for oral bioavailability were ten or fewer rotatable bonds (RBs) and a topological polar surface area (TPSA) not exceeding 140 Å^2^ [113]. However, data from Table 4 indicate that extended boundaries of up to 14 RBs and a TPSA of up to 200 Å^2^ are acceptable, which broadens the scope compared with the Veber guidelines. Additionally, when compared with the Rule of Five (Ro5), our results suggest a notably broader spectrum for molecular weight (MW) and calculated log P (clog P). According to our findings, molecules can reach an MW of 950 and have a clog P ranging from 1 to 7. While the Ro5/Veber guidelines indicate a nitrogen atom count (NAr) limit of 5 for favorable drug attributes, our data suggest a more pragmatic boundary would be an NAr of 4, except for ACBI2, which is a VHL-based PROTAC. Kihlberg has categorized the property space for small-molecule drugs into three distinct zones (the traditional Rule of Five (Ro5) in line with Lipinski’s criteria, the extended Rule of Five (eRo5) space, which reflects the broader scope within the conventional Ro5 distribution, and beyond Rule of Five (bRo5) territory, where additional mechanisms such as hydrophobic collapse and compensatory hydrogen bonding come into play to facilitate oral bioavailability) [114,115]. The softer boundaries for characteristics like clog P and TPSA could be attributed to the versatile nature of larger molecules, which can adapt their structure and, consequently, their effective physicochemical traits to suit their surroundings. In contrast, one encounters definite limits with hydrogen bond donors, which adhere more closely to the original Ro5 parameters. Prior research has highlighted HBDs as a primary barrier for oral absorption even within Ro5 confines, making this aspect the most limiting factor in the design of new compounds. Thus, these data suggest that, despite the expected challenges in achieving good oral bioavailability with bifunctional PROTAC molecules, the careful selection of ligands for the target protein, the employment of highly rigid linkers with good physicochemical properties, and the use of cereblon ligands can deliver orally active PROTAC degraders.

## 7. New Technology Based on PROTACs and Concluding Remarks

This brief review describes the advancements and challenges in the development of oral PROTACs over recent years as they progress through clinical trials. The concept of targeted protein degradation has surged in popularity due to its significant potential [116]. Although PROTACs demonstrate several advantages compared with traditional small-molecule inhibitors, they also exhibit similar drawbacks. PROTACs, derived from inhibitors of the protein of interest (POI), can still produce off-target effects. Additionally, their substantial molecular weight hampers cell membrane penetration and can lead to suboptimal pharmacokinetic profiles, diminishing their biological efficacy and therapeutic impact. Even though some PROTACs can effectively trigger protein degradation, their limited biological impact makes them less practical for disease treatment. Moreover, the scarcity of small-molecule binders for certain proteins, like transcription factors, which are pivotal in the onset and progression of many diseases, limits the breadth of PROTAC applications. It is critical to incorporate alternative drug design approaches into PROTAC technology to overcome these hurdles. In response to the challenges outlined, several innovative PROTAC-related technologies have emerged, such as Antibody-PROTACs [117], Aptamer-PROTACs [118], Dual-target PROTACs [119], Folate-caged PROTACs [120], and Transcription factor-PROTACs [121].

While these newer technologies offer distinct benefits over conventional PROTAC approaches, the development of oral PROTACs still faces a significant challenge. Our comprehension of traditional small-molecule ADME properties is deep and grounded in a broad, historical dataset, revealing a solid understanding of their physicochemical characteristics. By contrast, our understanding of PROTAC ADME attributes is still developing, with expectations that it will deepen as new data emerge. Predictive methods for human PK in small molecules are well established and have a high success rate. Whether these methods will apply directly to PROTACs, or if there will be a greater need for empirical data at the outset, remains to be seen. Despite initial worries about their physicochemical attributes, PROTACs have demonstrated ADME and PK profiles that are unexpectedly positive when contrasted with initial assumptions based on the Rule of Five. The forthcoming crucial hurdle for PROTACs as a therapeutic class will be to validate their pharmacological and PK properties in a clinical context, thereby paving the way to their successful development into efficacious drugs. Progress in launching these transformative treatments for patients will rely on the persistent efforts of scientists and the pharmaceutical industry to master the sophisticated endeavor of engaging with high-value proteins once deemed ‘undruggable’. Collectively, there exists the potential to forge a revolutionary path in the realm of cancer therapy, offering new hope for patients engaged in the fight against cancer.

## Figures and Tables

**Figure 1 pharmaceuticals-17-00494-f001:**
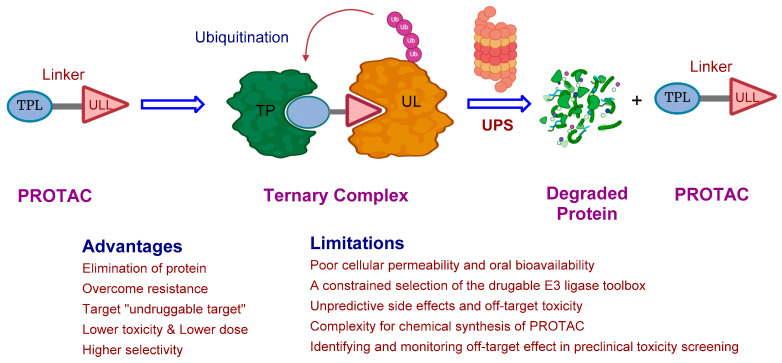
Mechanism of PROTAC-mediated protein degradation.

**Figure 2 pharmaceuticals-17-00494-f002:**
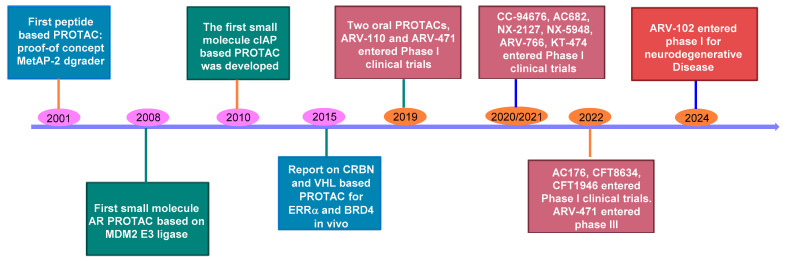
Development timeline of oral protein degraders in clinical trials.

**Figure 3 pharmaceuticals-17-00494-f003:**
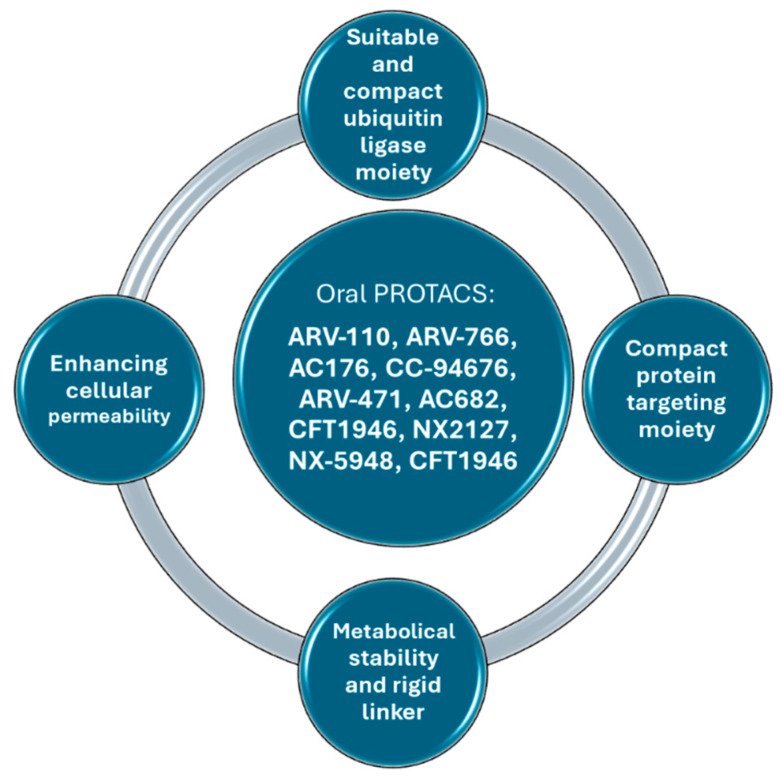
Required physicochemical properties for oral PROTACs.

**Figure 4 pharmaceuticals-17-00494-f004:**
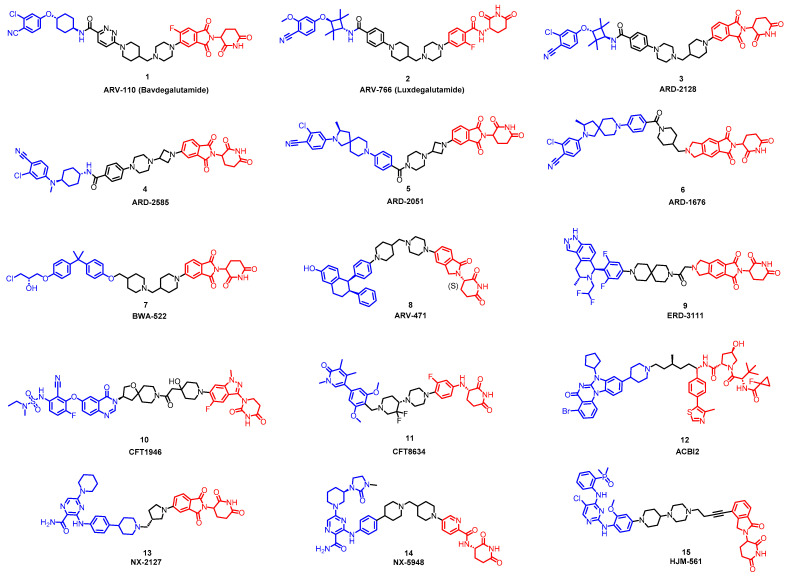
Representative orally bioavailable PROTACs for cancer.

**Figure 5 pharmaceuticals-17-00494-f005:**
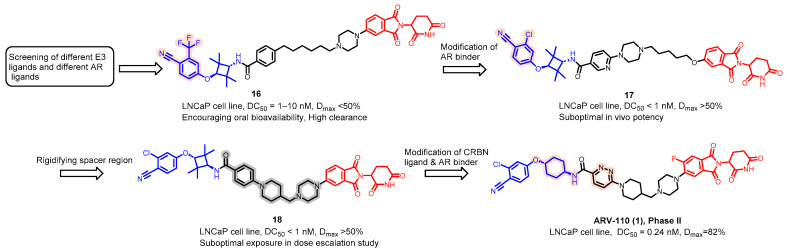
Medicinal chemistry campaign for the discovery of the potent and orally active AR degrader ARV-110.

**Figure 6 pharmaceuticals-17-00494-f006:**
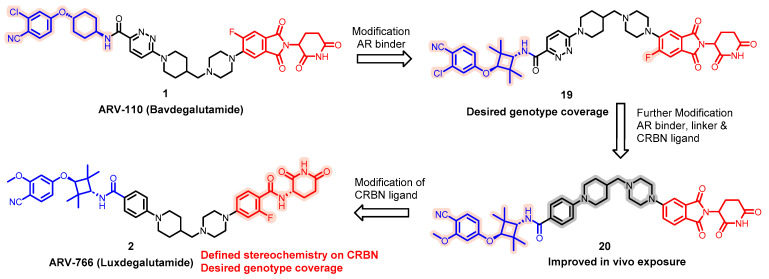
Discoveries along the way to ARV-766.

**Figure 7 pharmaceuticals-17-00494-f007:**
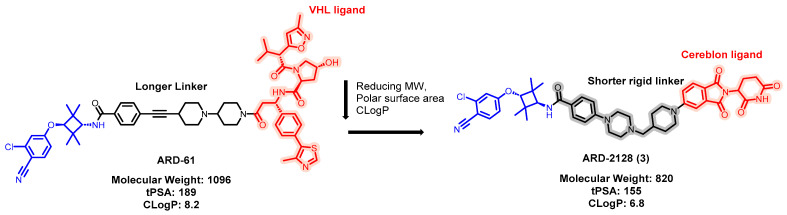
Medicinal chemistry campaign for the discovery of the potent and orally active AR degrader ARD-2128.

**Figure 8 pharmaceuticals-17-00494-f008:**
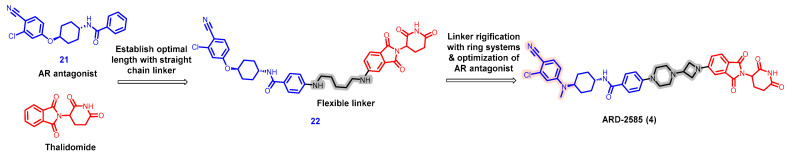
Medicinal chemistry campaign for the discovery of the potent and orally active AR degrader ARD-2585.

**Figure 9 pharmaceuticals-17-00494-f009:**
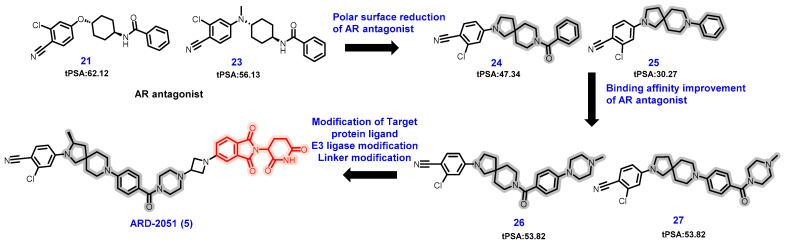
Medicinal chemistry campaign for the discovery of the potent and orally active AR degrader ARD-2051.

**Figure 10 pharmaceuticals-17-00494-f010:**
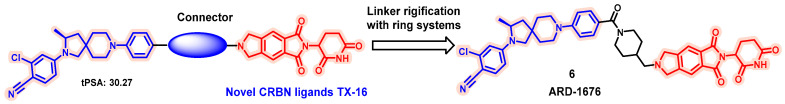
Medicinal chemistry campaign for the discovery of the potent and orally active AR degrader ARD-1676.

**Figure 11 pharmaceuticals-17-00494-f011:**
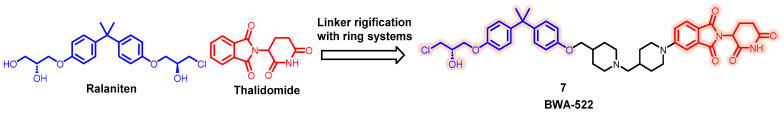
Medicinal chemistry campaign for the discovery of the potent and orally active AR degrader BWA-522.

**Figure 12 pharmaceuticals-17-00494-f012:**
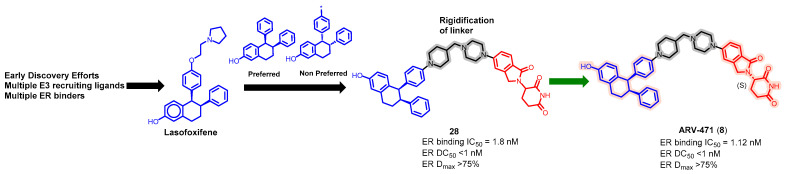
Medicinal chemistry campaign for the discovery of the potent and orally active ER degrader ARV-471.

**Figure 13 pharmaceuticals-17-00494-f013:**
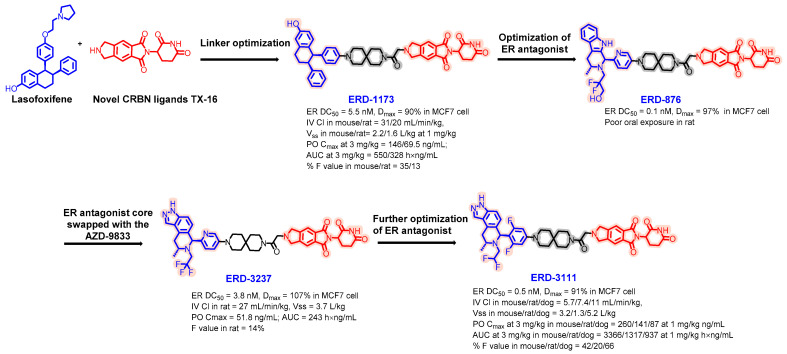
Medicinal chemistry campaign for the discovery of the potent and orally active ER degrader ERD-3111.

**Figure 14 pharmaceuticals-17-00494-f014:**
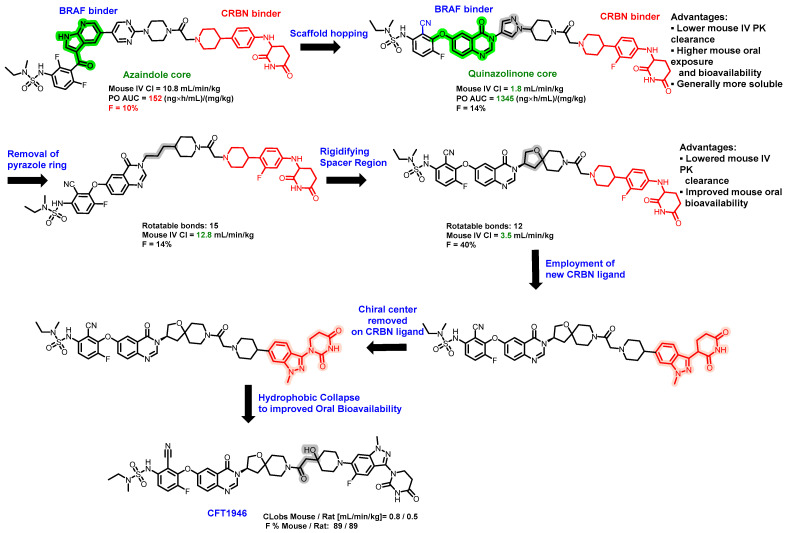
Medicinal chemistry campaign for the discovery of the potent and orally active BRAF degrader CFT1946.

**Figure 15 pharmaceuticals-17-00494-f015:**
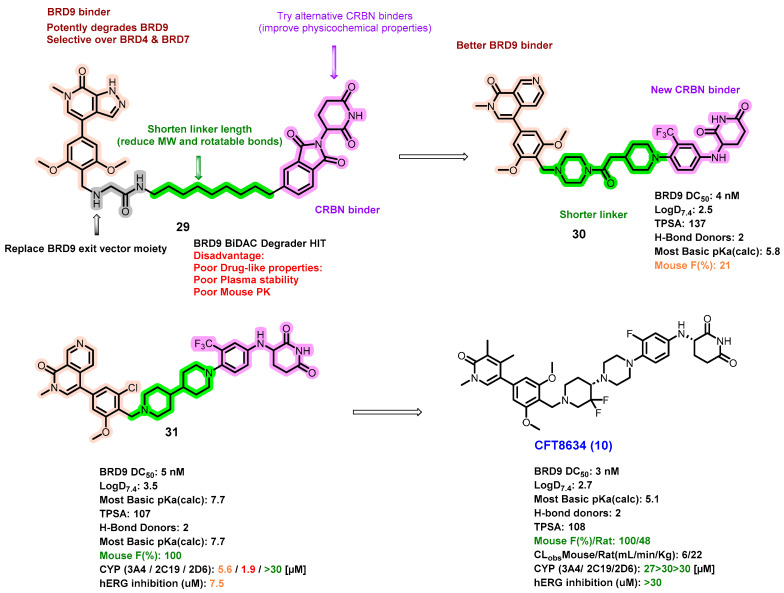
Medicinal chemistry campaign for the discovery of the potent and orally active BRD9 degrader CFT8634.

**Figure 16 pharmaceuticals-17-00494-f016:**
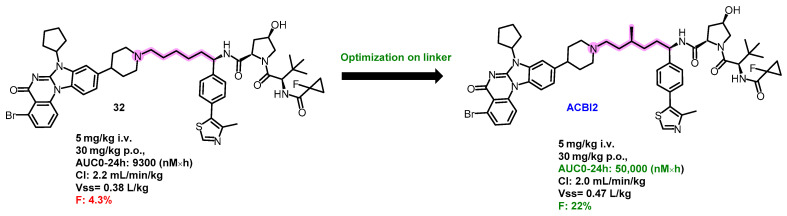
Medicinal chemistry campaign for the discovery of the potent and orally active SMARCA degrader ACB12.

**Figure 17 pharmaceuticals-17-00494-f017:**
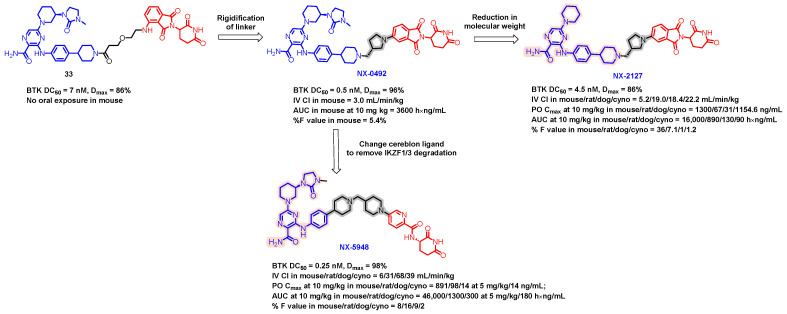
Medicinal chemistry campaign for the discovery of the potent and orally active BTK degrader.

**Figure 18 pharmaceuticals-17-00494-f018:**
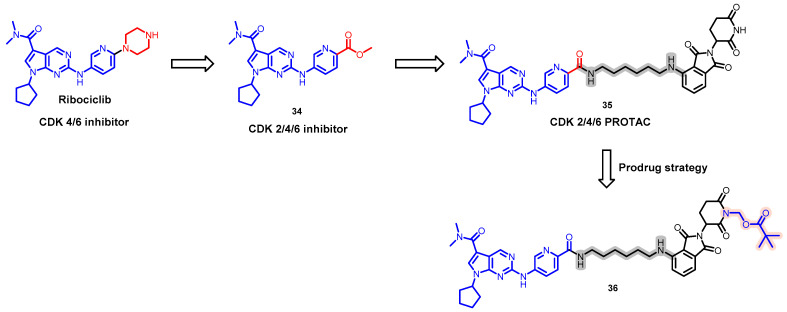
Medicinal chemistry campaign for the discovery of the potent and orally active CDK 2/4/6 degrader.

**Figure 19 pharmaceuticals-17-00494-f019:**
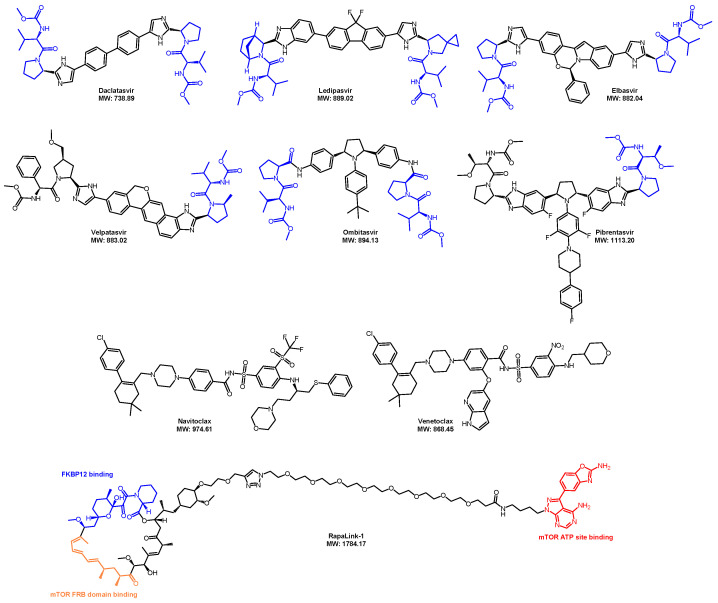
Representative example of an exception to Ro5 inhibitors.

**Table 1 pharmaceuticals-17-00494-t001:** List of Orally Bioavailable PROTACs in Clinical Development.

Degrader	Target	Title	Conditions	Clinical Trial Number (Status)
ARV-110	AR	Trial of ARV-110 and Abiraterone in Patients With Metastatic Castration-Resistant Prostate Cancer (mCRPC)	Prostate Cancer Metastatic	NCT05177042 (Phase I)
		Trial of ARV-110 in Patients With Metastatic Castration-Resistant Prostate Cancer	Prostate Cancer Metastatic	NCT03888612 (Phase II)
ARV-766	AR	A Study of ARV-766 Given by Mouth in Men With Metastatic Prostate Cancer	Prostate Cancer Metastatic	NCT05067140 (Phase II)
AC176	AR	A Study of AC176 for the Treatment of Metastatic Castration-Resistant Prostate Cancer	Metastatic Castration-Resistant Prostate Cancer	NCT05241613 (Phase I)
CC-94676	AR	Study to Evaluate the Safety and Tolerability of CC-94676 in Participants With Metastatic Castration-Resistant Prostate Cancer	Prostatic Neoplasms	NCT04428788 (Phase I)
ARV-471	ER	A Phase 1/2 Trial of ARV-471 Alone and in Combination With Palbociclib (IBRANCE^®^) in Patients With ER+/HER2− Locally Advanced or Metastatic Breast Cancer	Breast Cancer	NCT04072952 (Phase I/II)
		ARV-471 in Combination With Everolimus for the Treatment of Advanced or Metastatic ER+, HER2− Breast Cancer	Breast Cancer	NCT05501769 (Phase 1)
		A Study of ARV-471 (PF-07850327) Plus Palbociclib Versus Letrozole Plus Palbociclib in Participants With Estrogen Receptor Positive, Human Epidermal Growth Factor Negative Advanced Breast Cancer	Breast Cancer	NCT05909397 (Phase 3)
		A Study to Learn About a New Medicine Called ARV-471 (PF-07850327) in People Who Have Advanced Metastatic Breast Cancer.	Advanced Breast Cancer	NCT05654623 (Phase 3)
AC682	ER	A Study of AC682 for the Treatment of Locally Advanced or Metastatic ER+ Breast Cancer	Breast Cancer	NCT05080842 (Phase 1)
NX-2127	BTK	A Study of NX-2127 in Adults With Relapsed/Refractory B-cell Malignancies	B cell malignancies	NCT04830137 (Phase I)
NX-5948	BTK	A Study of NX-5948 in Adults With Relapsed/Refractory B-cell Malignancies	B cell malignancies	NCT05131022 (Phase I)
CFT8634	BRD9	A Study to Assess the Safety and Tolerability of CFT8634 in Locally Advanced or Metastatic SMARCB1-Perturbed Cancers, Including Synovial Sarcoma and SMARCB1-Null Tumors	Synovial Sarcoma, Soft Tissue Sarcoma	NCT05355753 (Phase I)
CFT1946	BRAF^V600E^	A Study to Characterize the Safety, Tolerability, and Preliminary Efficacy of CFT1946 as Monotherapy and in Combination With Trametinib in Subjects With BRAF^V600^ Mutant Solid Tumors	Solid Tumors, Melanoma, NSCLC	NCT05668585 (Phase 1/II)

**Table 2 pharmaceuticals-17-00494-t002:** The Prevalence of AR LBD Mutations.

Estimates of AR LBD Mutation Prevalence			
AR LBD Mutation	2016	2020	2023
L702H	~2%	~9%	~11%
T878X	~6%	~6%	~8%
H875Y	~4%	~4%	~5%

**Table 3 pharmaceuticals-17-00494-t003:** Profile Comparison between ARV-766 and Bavdegalutamide.

Potential to Improve Outcomes in Patients with Prostate Cancer	Bavdegalutamide	ARV-766
Degrades wild type and amplified AR	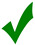	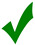
Targets all AR LBD mutations	No L702H	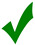
Tolerability suitable for mCRPC and mCSPC	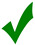	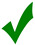
PSA50 in patients with tumors harboring the L702H mutation	7% (2 of 24)	50% (4 of 8)
Addressable mCRPC patient population	~11,000 (6–9%)	~35,000 (~25%)

**Table 4 pharmaceuticals-17-00494-t004:** The Physiochemical Properties of Currently Reported Orally Bioavailable PROTAC Degraders.

ID	Oral Degraders	MW	HBD	HBA	RB	TPSA	NAr	CLogP	Fraction Csp^3^
ARV-110	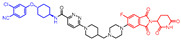	812	2	11	10	181	3	4.3	0.46
ARV-766	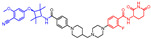	807	3	9	13	156	3	5.9	0.49
ARD-2128	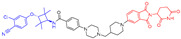	820	2	8	10	155	3	6.8	0.47
ARD-2585	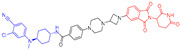	763	2	7	9	149	3	4.7	0.41
ARD-2051	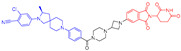	789	1	7	7	140	3	4.7	0.44
ARD-1676	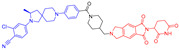	788	1	7	7	137	3	4.1	0.45
BWA-522	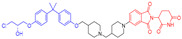	770	2	8	13	128	3	5.9	0.49
ARV-471	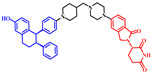	724	2	5	7	96	4	6.7	0.40
ERD-3111	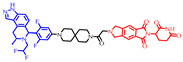	854	2	12	8	142	4	4.8	0.47
CFT1946	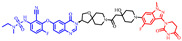	957	3	15	12	245	4	-0.55	0.44
CFT8634	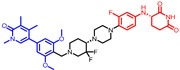	710	2	10	9	108	3	4.5	0.49
ACBI2	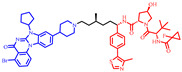	1064	3	9	19	182	5	10.3	0.54
NX-2127	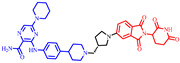	719	3	8	9	174	3	4.4	0.46
NX-5918	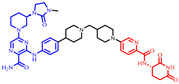	807	4	9	12	202	3	3.6	0.52
HJM-561	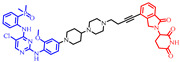	864	3	9	11	151	4	2.1	0.40
Ro5/Veber		500	5	10	10	140	ND	5	ND
bRo5		1000	6	15	20	250	ND	7.5	ND

## Data Availability

Data sharing is not applicable.

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
