# Peer review of "Orally Bioavailable Proteolysis-Targeting Chimeras: An Innovative Approach in the Golden Era of Discovering Small-Molecule Cancer Drugs"

_pharmaceuticals, 2024, doi:10.3390/ph17040494_

Round 1

Reviewer 1 Report

Comments and Suggestions for Authors

The Review entitled “Orally bioavailable PROTACs an innovative approach in the golden era of discovering small molecule cancer drugs” will be of interest not only to specialists in the field of medicinal chemistry, but to a wide range of scientists studying the fundamental mechanisms of action of small molecules on cancer cells of organisms.

The review is recommended for publication

Author Response

Thank you for the review and recommendation.

Reviewer 2 Report

Comments and Suggestions for Authors

The Authors conducted a systemic review regarding the latest advancements in orally delivered PROTACs (proteolysis-targeting chimeras) with potential in oncology that have entered clinical trials. The topic is interesting and the manuscript is well-written. The manuscript contains up-to-date information contained in easy-to-read tables. I believe that the manuscript is suitable for publication in Pharmaceuticals after editorial correction (including in the references list).

Author Response

(The authors gave the same response as above.)

Reviewer 3 Report

Comments and Suggestions for Authors

The manuscript summarized the development and optimization of orally bioavailable PROTACs (Proteolysis Targeting Chimeras) for cancer therapy, addressing the significant challenge of enhancing oral bioavailability despite the molecular complexities of PROTACs. It highlights the strategies employed to overcome the limitations set by the traditional "Rule of 5," including molecular engineering, optimization of physicochemical properties, and the innovative use of linker design and E3 ligase selection.

Some suggestions as follows that should be integrated to provide more depth, novel perspectives, and broader coverage of the topic:

1.      What do the authors what to present in figure 1? Why put a “rule of 5- for oral drug” here, is it closely relevant to PROTAC? One main thing the author discussed in the manuscript is that PROTACs are not fit the "Rule of 5”,  if you put this panel here, is it meanings "Rule of 5” is very important to PROTAC? Also there is potential plagiarism in this figure, the drawing and format of "Rule of 5” and “Advantages” are exact same as figure 2 in Cell Reports Physical Science 3, 101062, 2022.

2.      Figure 2 requires grammatical corrections and a clearer exposition to effectively communicate its intended message.

3.      The author should extend the topic of oral bioavailability of PROTACs, especially with the newly discovery of the role of IFITMs (interferon-induced transmembrane proteins) in facilitating the cellular uptake of various chemotypes, particularly PROTACs (Science, 2022). A. Emphasize the role of endogenous cellular proteins like IFITMs in aiding the transport of PROTACs across cell membranes. B. Highlight how the interaction with IFITMs or other similar cellular components could inform the optimization of PROTAC physicochemical properties, such as molecular weight, solubility, and linker length. C. Discuss the implications of these findings for the traditional "Rule of 5," reinforcing the idea that this rule may not fully apply to PROTACs.

4.      Including a discussion on innovative formulation techniques, such as nanoparticle delivery systems or solid dispersions, could offer valuable information on how to overcome solubility and stability challenges associated with PROTACs.

5.      While the author mentions the importance of linker optimization, further elaboration based on insights from the other papers could be beneficial. This could include discussing how linker length, flexibility, and composition affect the oral bioavailability, potency, and selectivity of PROTACs, supported by examples of successful linker modifications.

6.      Table 4, the bottom raw are misleading, what does the meaning of “Ro5/Veber”, “bRo5”, “PROTACs” and values meaning here?

7.      Emphasize the importance of ADME (Absorption, Distribution, Metabolism, and Excretion) studies in PROTAC development. A dedicated discussion on the role of ADME studies in predicting and improving the oral bioavailability of PROTACs, including methodologies and predictive models, could enhance the manuscript's comprehensiveness.

8.      Add future perspectives, such as next-generation PROTACs with improved properties; integration with other therapeutic strategies (e.g., immunotherapy, targeted therapy), etc.

9.      Add an abbreviation list.

Comments on the Quality of English Language

Most are ok, some grammatical errors can be found, especially in figure 2.

Author Response

Thank you for you suggestions and review. Please find the point to point reply in the attached word file.

Round 2

Reviewer 3 Report

Comments and Suggestions for Authors

Overall is good. the authors nicely addressed the reviewer's comments. Just please explain what is "tpl" "ull" "ul" in fig.1 caption.

Author Response

Thank you for the revision. The abbreviation for "tpl" "ull" "ul"  are added in the abbreviation list.